# Polymorphic design of DNA origami structures through mechanical control of modular components

Chanseok Lee [1], Jae Young Lee [1] & Do-Nyun Kim [1,2]

Scaffolded DNA origami enables the bottom-up fabrication of diverse DNA nanostructures by designing hundreds of staple strands, comprised of complementary sequences to the specific binding locations of a scaffold strand. Despite its exceptionally high design flexibility, poor reusability of staples has been one of the major hurdles to fabricate assorted DNA constructs in an effective way. Here we provide a rational module-based design approach to create distinct bent shapes with controllable geometries and flexibilities from a single, reference set of staples. By revising the staple connectivity within the desired module, we can control the location, stiffness, and included angle of hinges precisely, enabling the construction of dozens of single- or multiple-hinge structures with the replacement of staple strands up to 12.8% only. Our design approach, combined with computational shape prediction and analysis, can provide a versatile and cost-effective procedure in the design of DNA origami shapes with stiffness-tunable units.

[1] Department of Mechanical and Aerospace Engineering, Seoul National University, 301-dong 116-ho, 1 Gwanak-ro, Gwanak-gu, Seoul 08826, Korea. [2] Institute of Advanced Machines and Design, Seoul National University, 313-dong 320-ho, 1 Gwanak-ro, Gwanak-gu, Seoul 08826, Korea. Correspondence and requests for materials should be addressed to D.-N.K. (email: dnkim@snu.ac.kr)

The extensive design space of scaffolded DNA origami[1–3] comes from the availability of drawing a unique scaffold pathway with corresponding sequence design of staple strands for each structure. By utilizing that, a number of different structures were created including 2D planar sheets[1,4], 3D bundle structures with various shapes and curved forms[5,6], polyhedra[7,8], and wireframe-based assemblies with complex geometry[9,10]. Also, there has been many attempts to construct dynamic structures whose direction and range of motion can be programmed[11] in order to be used as kinematic components[11,12], high-resolution positioners[13,14], mechanical testing units[14–18], and reconfigurable structures operated by fuel strands or external stimuli[19–24].

To make them, the scaffold pathway, determined by the cross-section shape of DNA bundles and the layout of scaffold crossovers, is a primary design parameter. For structures with curved or flexible regions used as vertices or rotational joints, the modification of scaffold pathway at those regions has been adopted so far to change the helicity of DNA bundles[6] or to reduce the number of DNA helices at the cross-section[7,8,11–14,18,21]. A disadvantage of this approach is that a designer has to replace a large number of staple strands when the design is in need of revision even slightly, because the modification of the pre-determined scaffold pathway induces the sequence alteration of related staples even though they remained at the same position in the structure. It has been considered as an innate and inevitable limitation of using a long scaffold with pre-determined sequence. Therefore, modular design approaches, successfully employed in small-sized tiles or brick-based origami[25,26], have not yet been widely introduced in scaffolded DNA origami despite its usefulness for programming a wide range of variations in the bent shape and mechanical stiffness of the structure. A modular design method using two-dimensional repeating scaffold pathway to create two- and three-dimensional structures was only recently reported[27], though it excludes conventional lattice-packing designs[5] and has

limited ability to control mechanical stiffness of the module. Such limitation puts a significant burden in terms of cost and time in the laboratory-scale synthesis, design modification and optimization of various DNA origami structures, acting as a major obstacle to their widespread use in many related research fields. While some other approaches to reducing the fabrication cost were reported[28,29], that they require a custom scaffold for each structure.

Here we demonstrate polymorphic variation of the reference, 12-helix bundle design, by selective replacement of constituting staples in desired modules only. We find the controllable range of bending stiffness and included angle of the structure, as well as the folding characteristics depending on the structural complexity and rigidity. Since our design method is based on the conventional lattice-packing rule[5] and compatible with commonly used design program[30], it is expected to be adopted in DNA nanostructure field easily and yield diverse structural variations with a broad range of application.

## Results

**Design concept and requirements**. Our modular design method starts from partitioning the structure by drawing a periodic scaffold path filling the cross-section. A unit module region is sandwiched between two cross-sections consisting of aligned scaffold crossovers, named as seam regions (Fig. 1a). These scaffold seam regions play a central role in blocking the propagation of sequence alteration as well as in increasing the structural stability. Basically, a structure module is constructed by filling the helices in the module region with staples using hexagonal lattice-packing[5], and remains in rigid state. To make a flexible hinge module having rotational degree of freedom, we can simply eliminate the existing staples constituting the structure module while maintaining scaffold strands at the cross-section (Fig. 1b),

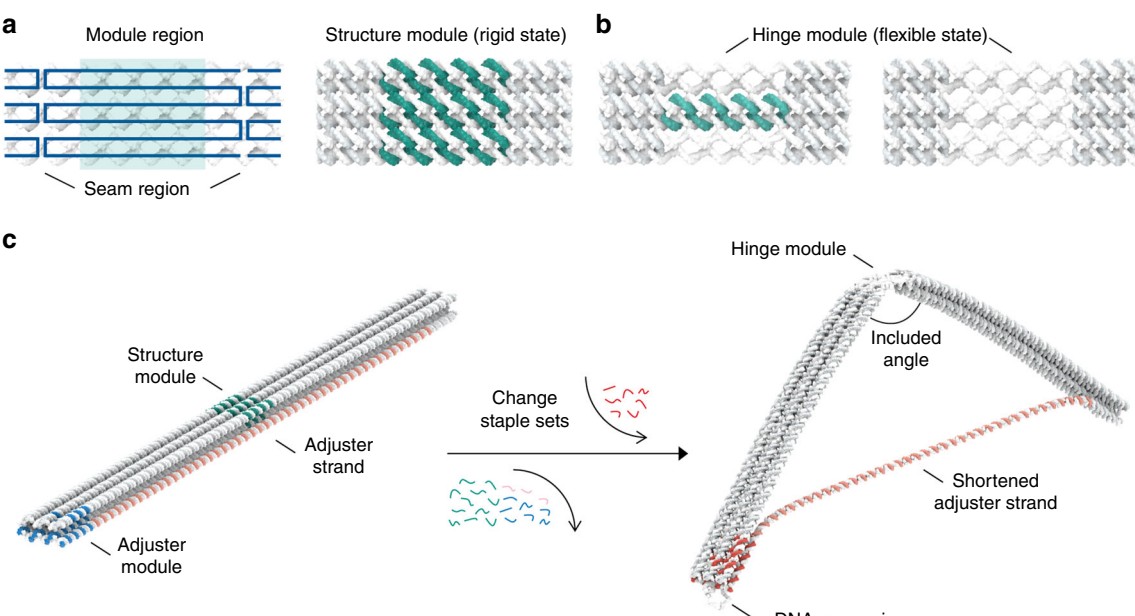

**Fig. 1** Schematic illustration of the modular design. **a** Scaffold strand is represented as blue lines (simplified to show its pathway clearly). Scaffold crossovers are aligned at the cross-section, and these sections are located at regular intervals. A module region, shown as green-shaded area, is located in between the scaffold crossover seams. Basically, all the helices of the structure module are double-stranded and all possible staple crossover positions are connected. **b** The hinge module can be made by removing the staples in the structure module, while unbound scaffold ssDNAs at the module remained at the cross-section. By controlling the number of dsDNA and staple crossovers, bending stiffness of the hinge module can be adjusted. Seam region maintained with full connection throughout the modification in order to ensure structural stability. **c** A 12-helix honeycomb-latticed bundle design to illustrate the method of controlling the shape of the structure. To make a bent structure from the straight one, staples of the structure module were removed and the adjuster module component was changed in order to make a shorter adjuster strand. Excessive strut staples were eliminated

instead of reducing the number of DNA helices as in the previous studies[7, 11–13, 21]. The bending stiffness of the hinge module can be tuned by controlling the number of dsDNA and staple crossovers inside the hinge module (Supplementary Fig. 1). We also incorporate a long scaffold helix passing through the body into our design as an adjuster strand[21,31], whose length can easily be varied by the adjuster module (Fig. 1c). As it shortens, the included angle of the structure is decreased, and the remaining part of the adjuster strand is stored in a reservoir at the end of the structure. The adjuster strand is basically formed as dsDNA by the aid of strut staples.

We found that both hinge module and adjuster strand were necessary to create a uniformly bent monomer structure as intended (Supplementary Figs. 1–3). Structures having a hinge module at the center of the body without an adjuster strand did not bend properly into the target shape. More than half of the monomer structures remained straight even though the most flexible hinge module was used (Supplementary Fig. 1). When the structure had an adjuster strand solely without any hinge module in the body, most structures formed aggregates rather than remained as a monomer (Supplementary Figs. 2 and 3). We could observe kinks developed at arbitrary positions, even if they were folded into monomeric structures. The seam spacing and the length of the module region seem to affect the folding quality and the spatial resolution of the shape slightly (Supplementary Figs. 4 and 5).

**Polymorphic shape variation from the reference design**. To demonstrate the efficiency and versatility of our module-based design method, we designed a reference structure consisting of 12 helices on the honeycomb lattice[5] that can provide polymorphic structures with minimal staple changes (Fig. 2 and Supplementary Figs. 6–13). The reference structure, consisting of 180 unique staple strands, was folded into a straight bundle as it did not have any hinge and a 504 nucleotide (nt)-long dsDNA adjuster (Supplementary Fig. 6). Seam regions divide it into nine module regions (L1, L2, L3, M1, M2, M3, R1, R2, and R3). Seven of them from L2 to R2 serve as potential locations for hinge modules and R3 region is used as the adjuster module (Supplementary Fig. 7 and Supplementary Table 1). To illustrate, we built 24 representative polymorphous constructs by revising the module designs while sharing most of the staples (Fig. 2). Here, all structures were designed to have planar shapes in order to be measured their geometrical features clearly by atomic force microscopy (AFM). CanDo modeling framework[3,32], highly utilized to design these structures prior to fabrication, provided the equilibrium folding shapes remarkably consistent with experimental observation (Supplementary Figs. 14–16).

The number of replaced staples, whose sequences are different from those of the staples in the reference pool, is 7 (3.9% of the reference staples) for single-hinged structures with various hinge module locations and included angles (cases 1 to 8), 7–11 (3.9 ~ 6.1%) for double- or triple-hinged structures with a single adjuster strand (cases 9 to 16), and 10–23 (5.6 ~ 12.8%) for more complex structures that are closed-form or with double/asymmetric adjuster strand(s) (cases 17 to 24) (Supplementary Table 2). Note that the number of staple replacement above is provided for each structure in comparison with the reference structure. In fact, when all the cases are considered, the actual number of staple replacement is much smaller since many replaced staples are shared in multiple structural variants (Supplementary Data). All constructs were folded successfully at high monomer folding yields ranging from 73.5 to 89.6% (Supplementary Fig. 17 and Supplementary Table 3). Structural yield obtained by counting the portion of correctly assembled structures among monomers from AFM images, on the

other hand, ranged from 32.6% (case 16) to 93.4% (case 5) (Supplementary Fig. 18 and Supplementary Table 4). In general, single-hinged structures showed relatively high structural folding yield (77.5 ~ 93.4%) compared with those of double- and triple-hinged structures (32.6 ~ 84.1%), which might originate from the existence of multiple stable configurations. Structures with multiple hinges modulated by a single adjuster might have several local energy minimum states and be trapped there during annealing process with complex assembly kinetics, leading to the lack of shape homogeneity. We could circumvent this issue by assigning an adjuster for each hinge separately, which nevertheless required an increased number of modified staples (Supplementary Fig. 19). Another factor affecting structural integrity was the proper arrangement of hinge position and adjuster length. When the hinge existed at a largely asymmetric position and the dsDNA adjuster was too short, the structure tended to be severely distorted and failed to be folded into a proper shape as predicted by CanDo analysis (Supplementary Fig. 20).

**Hinge angle variation**. In addition to the diverse shape variation, we can also control the included angle of an individual hinge module more precisely. To illustrate, we introduced a flexible hinge containing two dsDNA strands at the center (M2) module region and changed the length of the adjuster by 21 nt basis. Structures with the included angle ranging from 0° (folded in half) to 150° at an interval of 15° were successfully fabricated at a high monomer folding yield, and with structural integrity for the entire range (Fig. 3, Supplementary Figs. 21 and 22, and Supplementary Table 5). Even finer control of the included angle may be achieved by adopting a shorter basis in adjuster strand or placing the adjuster closer to the hinge while it narrows the controllable range of the included angle[12,13]. A noteworthy advantage of our design method is that only 7–10 new staples (3.9 ~ 5.6% of the number of reference staples) are required to control a wide range of included angles from the straight structure (Supplementary Table 2). This portion of substituted staples is significantly smaller than that reported in previous studies (~16[12] and ~30%[6]), demonstrating the excellent efficiency of the proposed method in realizing polymorphic DNA origami designs. Angle distribution became relatively wider for the target included angle greater than 120° (Fig. 3c), which might originate from high variability of the hinge stiffness due to unbound scaffold ssDNA portions at the hinge module. The average end-to-end length of hinge ssDNA portions increased with the included angle, which elevated their entropic tensional force as predicted by wormlike chain (WLC) model[33,34] and the possibility of non-specific interactions among them, making the hinge stiffness more variable and less predictable. CanDo predictions supported our inference to some extent, because experimentally measured value of included angles lay between the value predicted by modeling ssDNA portions as non-interacting entropic springs and the value calculated by excluding them entirely from the model (Fig. 3d and Supplementary Figs. 23 and 24). Hence, it is suggested that multiple ssDNA portions at the hinge module do not fully provide tensional forces expected from an ideal model due probably to some interactions between adjacent strands.

In order to investigate the existence and frequency of non-specific interactions among single strands, we performed the MD simulation for structures with and without a hinge (Fig. 4 and Supplementary Figs. 25 and 26). The hinged structure has twelve 42-nt-long scaffold ssDNA portions at the center region (termed ds0hb hinge), whose cross-sectional shape is the same as those of the M2 module in the reference design. Since we simulated a part of the structure only near the hinge without the adjuster (Fig. 4a), MD simulation results might reflect the real condition better as

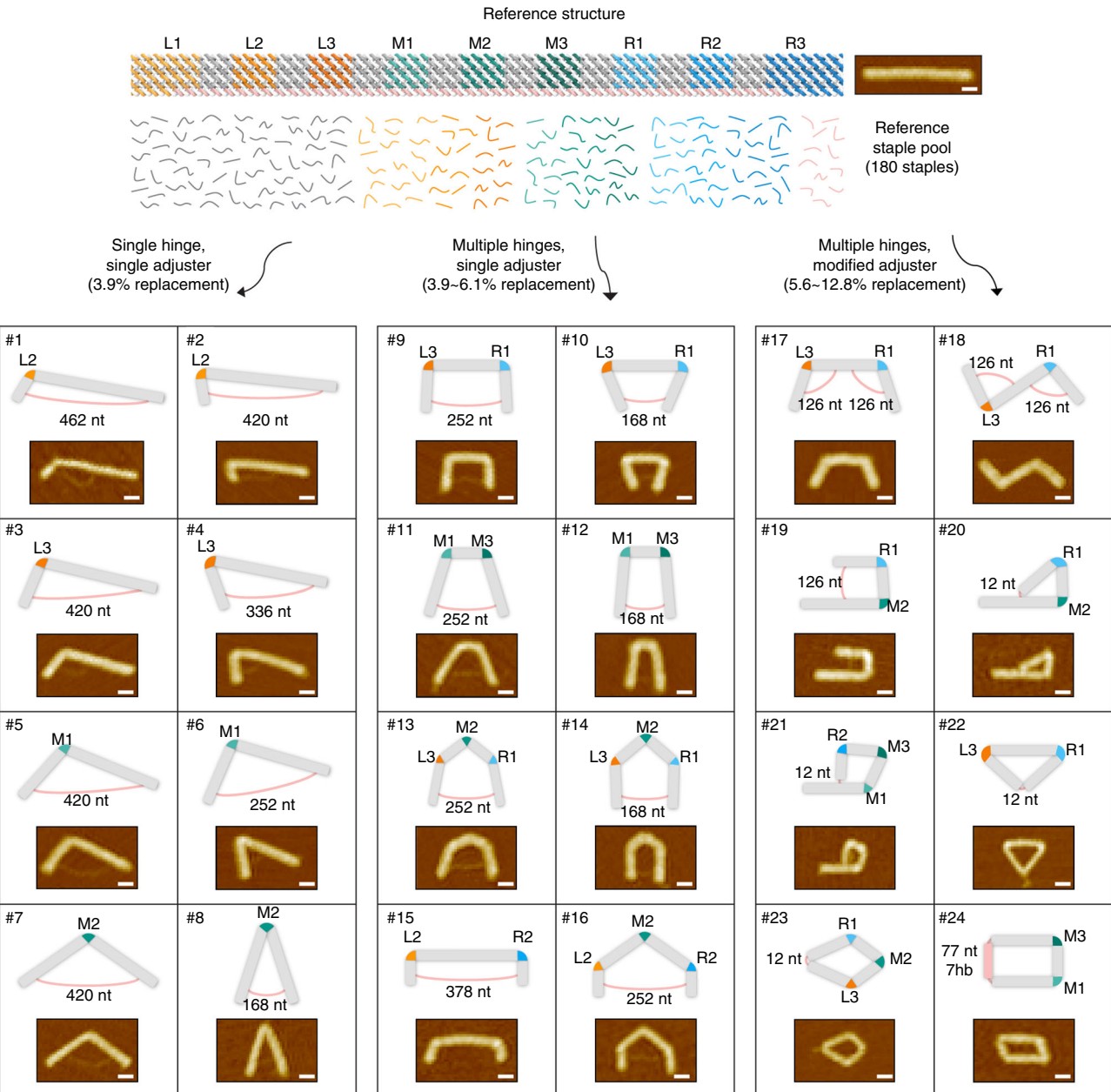

**Fig. 2** Demonstration of the 24 structural variations from the straight 12-helix honeycomb-latticed reference structure. The length of each seam region is about 28 nt (~9.5 nm), structural module L1 and R3 are 63 nt (21.4 nm), and rest of the structural modules are 35 ~ 42 nt (12 ~ 14.3 nm) long. In total, 180 staples constitute the reference structure consisting of 108 body staples, 60 seam staples, and 12 strut staples. See Supplementary Figs. 6 and 7 for AFM images and detailed layout of the reference structure, Supplementary Figs. 8–13 for large-area AFM images of each design variation, and Supplementary Figs. 14–16 for CanDo shape prediction. Scale bars: 30 nm

the included angle becomes closer to the straight conformation. We could confirm from MD trajectories that the hinged structure had higher fluctuation than the non-hinged one, and multiple non-specific interactions among ssDNA helices existed at the hinge module (Figs. 4b–d, and Supplementary Figs. 25, 26). On average, 8.1% of base pairing between bases in either same or different strands was observed in the hinge during equilibrium states from 90 to 110 ns (Fig. 4d).

**Effect of the hinge stiffness on folding characteristics**. To further explore the applicable range and value of the hinge stiffness, we developed five different hinge designs by varying the number of dsDNA portions and Holliday junctions at the M2 module

(Fig. 5a). The most flexible case was a ds0hb hinge, where all 11 structural staples were removed from the M2 module so that only unbound scaffold single strands remained. Stiffer hinges were then devised by adding 2–6 hinge staples to it leading to ds2hb, ds3hb, ds4hb, and ds6hb hinge structures, where each number indicates the number of dsDNA helices at the cross-section. Results from agarose gel electrophoresis and AFM imaging indicated that stiffening the hinge deteriorated the monomer folding yield and increased the number of aggregated structures (Fig. 5b, c, and Supplementary Fig. 27). Intensity ratios of the monomer band to all bands were 85.8, 79.1, 69.0, 31.1, and 7.0% for ds0hb, ds2hb, ds3hb, ds4hb, and ds6hb hinge, respectively. A drastic drop in the number of monomeric structures were observed for structures with ds4hb hinge and stiffer ones,

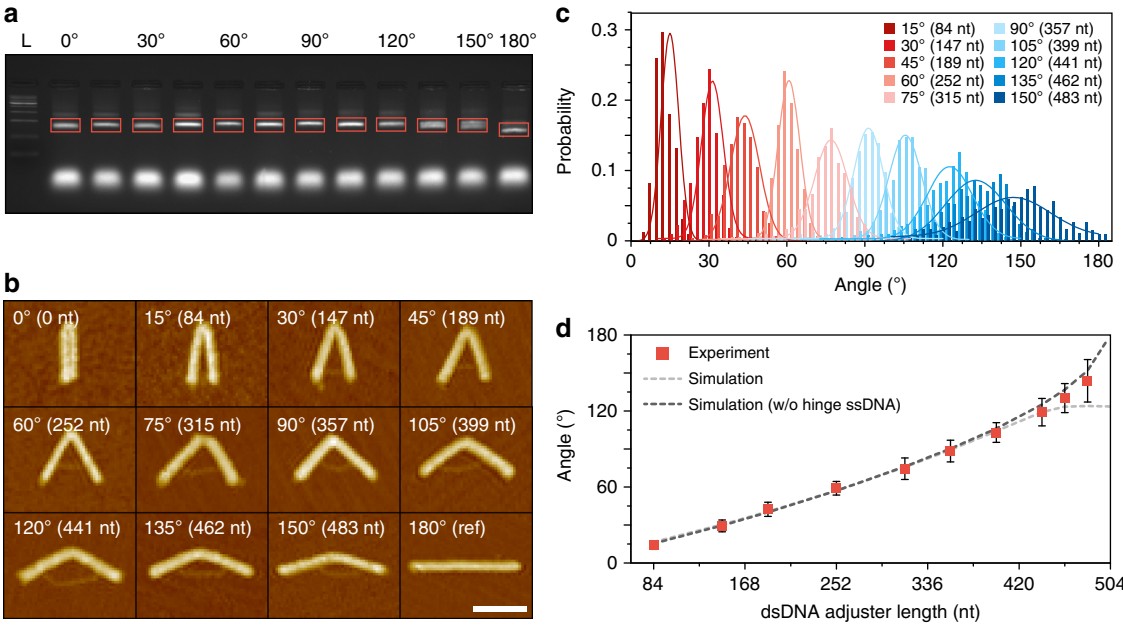

**Fig. 3** Broad and precise included angle variation of the hinge structure by controlling the length of the dsDNA adjuster strand. **a** Agarose gel electrophoresis result of each design. Orange boxes are monomer structure bands, and the bottom bands are excessive staples. L: 1 kb DNA ladder. **b** Representative AFM images of each structure. Large-area AFM images of each structure were shown in Supplementary Figs. 6, 21 and 22. Scale bar: 100 nm. **c** Histograms of included angle distribution of each structure. Solid lines indicate the Gaussian distribution. See Supplementary Table 5 for detailed results. **d** Measured average included angle and CanDo analysis result. Error bars indicate standard deviation of experimental data. See Supplementary Figs. 23-24 for the CanDo results

recommending the use of a hinge softer than ds4hb hinge in practice.

We quantified the stiffness of ds0hb, ds2hb, and ds3hb hinges, by adopting a ssDNA adjuster whose tensional force was modulated through a systematic variation of their length[15, 31] and measuring the included angles (Fig. 5d and Supplementary Figs. 28–31). The hinge stiffness was estimated by modeling the hinge as a torsional spring under the tensional force from the adjuster calculated using the WLC model (Supplementary Methods). The estimated values of bending stiffness on average were 25.3, 33.8, and 49.6 pN nm rad$^{-1}$ for ds0hb, ds2hb, and ds3hb hinge, respectively (Fig. 5e). Principal component analysis (PCA) of MD trajectories obtained for these hinge modules showed a similar level of stiffening of the hinge with the increase of double-stranded portions (Supplementary Figs. 32–35), although the absolute stiffness values were different from those estimated from AFM images as only the hinge part was simulated in MD. Strong binding of staple strands to the scaffold strand at the hinge was also observed throughout the simulation, demonstrating that the stability of our hinge designs with controllable stiffness (Supplementary Fig. 36).

Using the estimated hinge stiffness, we calculated the total strain energy of each hinge design as a function of included angle by summing the strain energies in the hinge module and the ssDNA adjuster assuming negligible deformation in other modules (Fig. 5f and Supplementary Fig. 37). The mean included angle determined experimentally coincided with the value where the total strain energy became minimum[35]. As the hinge got stiffer, the minimum strain energy increased naturally, and the strain energy became more concentrated in the hinge module (Supplementary Fig. 38), which partly explained the deterioration of the folding yield of monomeric structures with stiff hinges. Hinge staples, when the designed hinge stiffness was too high, might tend to be bound to other neighboring structures to form less-bent multimeric structures, which would be energetically

more favorable than the formation of monomeric hinge structures as observed in AFM images (Fig. 5c). For structures with the dsDNA adjuster, the mean included angle was almost independent of the hinge stiffness, whereas it was still dependent on the length of the adjuster (Supplementary Figs. 39–42). Also, their included angles were much less deviated from the mean value compared to the structures adjusted by ssDNA. The mean and deviation of the included angle could be more-finely tuned by simply adding a few strut staples binding to the adjuster and controlling the portion of dsDNA (Supplementary Figs. 43–45). Therefore, it offers a versatile way of programming the target mean angle and flexibility of hinge structures statically and also dynamically through addition or removal of required staples[19, 24].

## Discussion

In summary, our module-based design method provides an efficient way to control the local stiffness of the DNA origami structures, thereby expanding the design space even with a highly limited range of replaced staple sequences. While we demonstrate our design approach here for a honeycomb-latticed bundle structure, the same design principle can be easily applied to other types of structures including planar sheets[1, 19] and bundles with various cross-section shapes and packing rules[36]. By adopting our method, one can test a wide range of geometrical variations in a highly cost-effective manner, to utilize it as a design platform by placing various functional nanoparticles in desired position and orientation. In addition, by changing the adjuster from a scaffold strand to the fuel or functionally modified strand, our design can be directly utilized to the dynamic mechanical component driven by external stimuli while the range of motion can be controlled by the stiffness of the hinge. Also, CanDo analysis can be used as pre-screening and validation of the shape and feasibility of the structure before fabrication, which leads to significant enhancement of the design efficiency in the scaffolded DNA origami. A deeper understanding of the effect of scaffold route design and

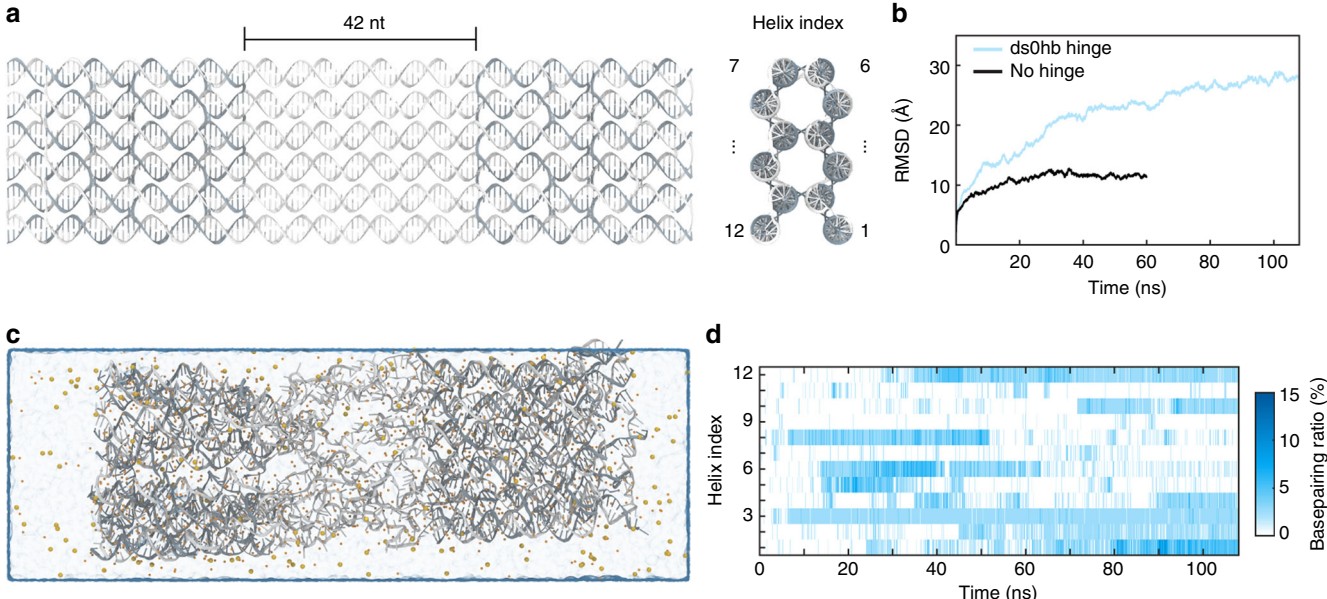

**Fig. 4** MD simulation result of the hinge module. **a** Initial configuration of the structure. **b** Root-mean-square deviation (RMSD) value during MD simulation. **c** A snapshot showing the configuration of the hinged structure at the final time step (after ~110 ns of simulation time). More snapshots of hinged and non-hinged structures were shown in Supplementary Figs. 25 and 26. **d** Base pairing ratio of each helix at the hinge module throughout the MD simulation. Base pairing can occur by the interaction of ssDNA bases between either same or neighboring helix

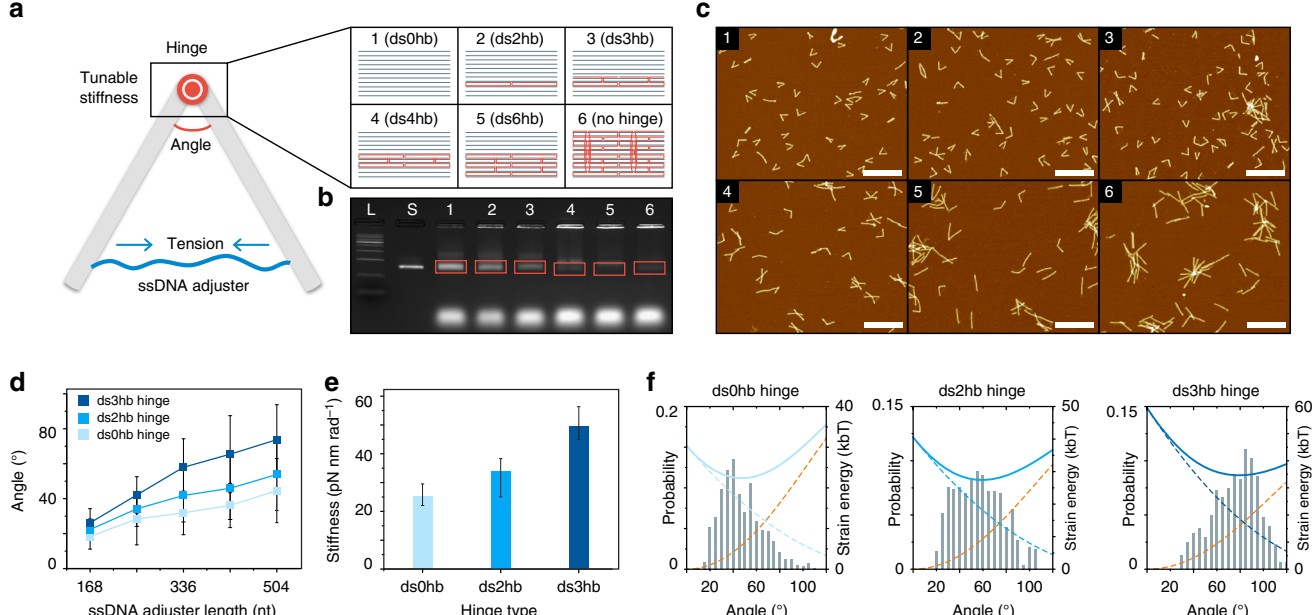

**Fig. 5** Analysis of the hinge stiffness. **a** Schematic illustration of the test design. Here, a ssDNA adjuster strand was used to make it as a tensional component. The stiffness of the hinge can be controlled by changing the number of dsDNA and staple crossovers at the hinge section. **b** Agarose gel electrophoresis result. Orange boxes are monomer structure bands, and the bottom bands are excessive staples. L: 1 kb DNA ladder. S: M13mp18 scaffold. 1–6: folded structures with each hinge stiffness. **c** AFM images of each hinge design. See Supplementary Fig. 27 for large-area images. Scale bars: 500 nm. **d** Measured average included angles of three different hinge designs. Error bars indicate standard deviation. See Supplementary Figs. 28–31 for more details. **e** Calculated bending stiffness of each hinge design. Error bars indicate maximum and minimum stiffness values. **f** Strain energy analysis of the structures with different hinge designs with a 504-nt-long ssDNA adjuster. Gray bars represent experimental included angle distribution, blue-dashed lines are the strain energy stored in the hinge, orange-dashed lines are the entropic energy of ssDNA adjuster, and the blue-solid lines indicate the summation of the two energies, respectively. See Supplementary Fig. 37 for a complete set

corresponding folding pathway of the structure during annealing process in the scaffolded DNA origami[37] may be useful to enhance the design efficiency and structural quality further.

## Methods

**Materials.** M13mp18 single-stranded DNA (7,249 nt length) was purchased from New England Biolabs (N4040s), and staple strands were provided by Bioneer Corporation (www.bioneer.co.kr). The list of all staple strands used in experiments are shown in Supplementary Data. DI water, TAE buffer, and $MgCl_2$ solution with molecular biology grade were purchased from Sigma-Aldrich.

**Design and assembly of DNA origami structures.** DNA origami structures were designed using caDNAno software[30] and CanDo[3, 32] modeling approach. Detailed modeling method about DNA origami structures and solution procedure are described in Supplementary Methods. The final folding mixture had 10 nM concentration of scaffold DNA, 100 nM of each staple strands, 1 × TAE buffer (40 mM Tris-acetate and 1 mM EDTA), and 20 mM of $MgCl_2$. For self-assembly process, the mixture was annealed with a temperature gradient from 80 to 65 °C by −0.25 °C per minute and 65 to 25 °C by −1 °C per hour in a thermocycler (T100, Bio-Rad).

**Agarose gel electrophoresis.** Folded DNA origami structures were electrophoresed using 1.5% agarose gels containing 0.5 × TBE (45 mM Tris-borate and 1 mM EDTA, Sigma-Aldrich), 12 mM $MgCl_2$, and 0.5 μg ml$^{-1}$ of ethidium bromide (EtBr, Noble Bioscience Inc.). Samples loaded in an agarose gel were allowed to migrate for 1.5 h at 75 V bias voltage (~3.7 V cm$^{-1}$) in an ice-water cooled chamber (i-Myrun, Cosmo Bio CO. LTD.). Gel imaging was performed using GelDoc XR+ device and Image Lab v5.1 program (Bio-Rad).

**AFM imaging.** To avoid the possibility of unintended deformation or change of mechanical properties of the DNA origami structures induced by EtBr intercalation and mechanical damaging during gel electrophoresis, only unpurified samples were used in AFM measurement. One microlitre of annealed sample was diluted using 19 μl of folding solution (1 × TAE, 20 mM $MgCl_2$), and deposited on a freshly cleaved mica substrate (highest grade V1 AFM Mica, Ted-Pella Inc.). After incubation for 5 min, the substrate was washed with DI water and gently dried by $N_2$ gun for 1 min. AFM images were taken by NX10 (Park Systems) using non-contact mode in SmartScan software. A PPP-NCHR probe having spring constant of 42 N m$^{-1}$ was used in the measurements (Nanosensors). Measured images were flattened with linear and quadratic order using XEI 4.1.0 program (Park Systems). Structural folding yield analysis and included angle measurement of DNA origami monomer structures from AFM images were done by custom scripts using MATLAB R2015b software (MathWorks Inc.). Typically, more than 250 monomer structures obtained from at least two different AFM images were used for each case.

**MD simulation.** The starting atomic structures of 12-helices DNA bundles were generated using caDNAno[30] and CanDo[32]. Each hinge structure was solvated in a rectangular box of the TIP3P water model[38] with approximately 160 Å × 470 Å × 110 Å and neutralized to reach an ion concentration of 20 mM $MgCl_2$. MD simulation was performed using the NAMD[39] with the CHARMM36 force field[40], periodic boundary conditions, and the integration time step of 2 fs. The van der Waals and short-range electrostatic potentials were calculated using a12 Å cut-off. The long-range electrostatic interactions were computed using the Particle Mesh Ewald scheme[41] with the grid size of 1 Å. The potential energy of each system was minimized using the conjugate gradient method. For principal component analysis (PCA), the equilibrium trajectories of each 20 ns was calculated under the NPT ensemble with constant temperature and pressure of 298 K, and 1 bar using a Langevin thermostat[39], and the Nosé–Hoover Langevin piston pressure scheme[42]. Calculating mechanical properties from PCA data is described in Supplementary Methods.

**Data availability.** The data that support the findings of this study are available from the corresponding author upon request. Computer code is available from GitHub at https://github.com/ChanseokLee/DNA_angletest.

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

## Acknowledgements

This work was supported by the National Research Foundation of Korea (NRF) grants funded by the Korea government (Ministry of Science and ICT) (NRF-2016R1C1B2011098, NRF-2017M3D1A1039422, and NRF-2014M3A6B3063711), as well as by Aspiring Researcher Program through Seoul National University (SNU) in 2014.

## Author contributions

C.L. and D.-N.K. conceived of modeling and design approach. C.L. performed the experiments and analyzed the data. J.Y.L. performed MD simulation and analyzed the data. C.L. and D.-N.K. discussed the results and wrote the manuscript. All authors commented on and edited the manuscript.

## Additional information

**Competing interests:** The authors declare no competing financial interests.

