## [Peer Review File · Nature Communications]

Reviewers' Comments:

Reviewer #1:

Remarks to the Author:

In their manuscript Lee et al. are presenting a method to bend a DNA origami rod into a subset of structures through a number of hinges that are selectively made flexible through the depletion of staple strands in the hinge region. To force bending around the hinge(s) a so-called adjuster duplex(es) is inserted to keep the structure in shape, much like the string on a bow (albeit restricting the bending to the hinge region). This is demonstrated through the formation of 24 different shapes of the origami rod as imaged by AFM.

It is a novel and interesting method to fold a DNA origami structure into different shapes by removing just a few staple strands. However, I have a some reservations listed below:

1) The method generally works well for structures where only one of the hinges is bent, however for structures where two or three hinges are bent yields are highly variable. As an example only a small fraction of structures with the expected shape are observed for #16 in fig. S10. The authors should define a set of rules for well-folded structures and then calculate the yields of well-folded structures from the AFM images.

2) The authors nicely demonstrated in Figure 5 that the angles can be precisely controlled by varying the length of the adjuster duplex. However these structures are static and it would have been interesting to include structures where the adjuster strand could be varied dynamically by strand displacement, i-motifs or other mechanical designs. Anyhow that may be the subject of future studies.

3) Some improvement of the language is required. (E.g. in the discussion line 2: "thereby to explode the design space" should probably be ""thereby to explore the design space"

I do have some concerns about the efficiency and yield of the method, however it does introduce a new design principle, and if the authors can address the issues mentioned above satisfactorily the manuscript may be come suitable for publication in Nat. Commun.

Reviewer #2:

Remarks to the Author:

In their manuscript "polymorphic design of DNA origami structures through mechanical subunit control", C. Lee and colleagues describe the design and construction of DNA structures with hinges that can be bent into different angles by the contracting force of an adjuster strand. Even though this idea itself is not new anymore (ref. 12, 15, 30...), they gave it an interesting and elegant new twist by introducing a modular design that allows them to convert an individual module into a hinge by omitting some staple strands. They studied their system thoroughly and show many examples with one or more hinges, modulated the hinge bending modulus, angles, and the adjuster strands and accompanied their experiments with MD and finite element simulations. They cite relevant literature and their analysis is sound. Their approach seems very useful and is likely to be picked up by other laboratories as it is easy to implement and is compatible with caDNAno, the standard design program in the field.

This reviewer therefore recommends publishing some minor revisions.

The authors do not define their nomenclature properly and it is not clear where they are interchangeable (body module / structure module region / structure module; module with reduced stiffness / hinge / hinge region / hinged region; adjuster module / adjuster strand / adjuster strut; strut staples / adjuster staples...). This should be better defined in the beginning of their design paragraph and the same terminology should be used consistently throughout the manuscript,

supplement and figures.

Specify in fig. 1b (or in a supplementary figure) how exactly a rigid unit is converted into a hinge.

Is it correct that staples are omitted or do in addition staples have to be exchanged?

The language should sometimes be more precise, for example some adjectives are seem misplaced.

- line 22 "unprecedented efficiency" sounds a bit bold and it is not clear what exactly is meant.

How about ...can provide a powerful, user-friendly and cost-effective design approach... or something similar?

- l. 26: What is an "extensive design domain" or l. 28 an "escalating demand"?

- l. 37: No, the sequence of the scaffold strand never changes. A different segment is hybridized to the adjuster module, right?

- l. 51: what is meant by sequence-independent region?

Other minor points:

- Title: The title could focus more on the modularity, which is in this reviewer's opinion the main selling point.

- l. 41: ref. 27 actually also reported 3D structures, not only 2D, but that approach is not suitable for the mechanical control Lee et al. show here.

- Past and present tense is mixed up repeatedly, e.g. in the results or l. 242 (was instead of is changed)

- There should always be a space between number and unit (often nt is used without space in text and figures)

- Figure 2: delete "~" in front of numbers. Is 12.8% replacement not accurate? Is replacement actually correct or are some staple strands just omitted?

Response to the referees' comments

We appreciate the reviewers' thorough reading of our manuscript and useful comments. Our point-by-point response follows:

Reviewer #1's comments to the authors:

(1) The method generally works well for structures where only one of the hinges is bent, however for structures where two or three hinges are bent yields are highly variable. As an example only a small fraction of structures with the expected shape are observed for #16 in fig. S10.

The authors should define a set of rules for well-folded structures and then calculate the yields of well-folded structures from the AFM images.

► As the reviewer pointed out, analysis on the structural integrity is important in order to validate the performance of our design method and develop it further. Therefore, we performed quantitative analysis on the structural folding yield of each design shown in Fig. 2, and we added a description on the result in the main text with Supplementary Figs. 18 and 19.

“Structural yield obtained by counting the portion of correctly assembled structures among monomers from AFM images, on the other hand, ranged from 32.6% (case 16) to 93.4% (case 5) (Supplementary Figs. 18 and 19). In general, single-hinged structures showed relatively high structural folding yield (77.5~93.4%) compared with double- and triple-hinged structures (32.6~84.1%), which might originate from the existence of multiple stable configurations.”

(2) The authors nicely demonstrated in Figure 5 that the angles can be precisely controlled by varying the length of the adjuster duplex. However these structures are static and it would have been interesting to include structures where the adjuster strand could be varied dynamically by strand displacement, i-motifs or other mechanical designs. Anyhow that may be the subject of future studies.

► The authors agree with the reviewer's opinion that introducing dynamic actuation to our design method by adopting a proper actuation mechanism would be highly interesting and widen the usability of our structure. In fact, we demonstrated the possibility of controlling the included angle by adding the strut staples in the Results section and Supplementary Figs. 45-47. We are currently investigating an efficient actuation mechanism triggered by a specific external stimulus (e.g., UV light, pH, fueling strand, etc.) applicable to our designs. However, the results are still preliminary and, therefore, we would like to leave it as future studies in this paper.

(3) Some improvement of the language is required. (E.g. in the discussion line 2: "thereby to explode the design space" should probably be "thereby to explore the design space"

► We revised that sentence following the reviewer's suggestion as *“thereby to expand the*

design space". Also, we revised some words and sentences of the manuscript for language improvement.

(4) *I do have some concerns about the efficiency and yield of the method, however it does introduce a new design principle, and if the authors can address the issues mentioned above satisfactorily the manuscript may be come suitable for publication in Nat. Commun.*

► The authors greatly appreciate the reviewer's comments. We hope that we properly addressed them in the revised manuscript where the results of quantitative structural folding yield analysis and a discussion on the folding efficiency are newly added.

Reviewer #2's comments to the authors:

(1) *This reviewer therefore recommends publishing some minor revisions.*

The authors do not define their nomenclature properly and it is not clear where they are interchangeable (body module / structure module region / structure module; module with reduced stiffness / hinge / hinge region / hinged region; adjuster module / adjuster strand / adjuster strut; strut staples / adjuster staples...). This should be better defined in the beginning of their design paragraph and the same terminology should be used consistently throughout the manuscript, supplement and figures.

► To avoid any confusion, we re-established terminologies (e.g., module region, structure module, hinge module, adjuster module, adjuster strand, and strut staples) and use them consistently throughout the manuscript and Supplementary Information.

- "*Module region*" indicates the section that can be formed either structure, hinge, or adjuster module depending on its location and constituting staple strands. It replaces "*body module*" and "*structure module region*" in the previous manuscript.
- "*Structure module*" is used to describe the module in rigid state. The term "*structure module region*" is not used anymore.
- "*Hinge module*" is used to describe the module in flexible state, replacing the term "*module with reduced stiffness*" and "*hinge(d) region*".
- "*Hinged structure*" is used when the structure has one or multiple hinges.
- "*Adjuster strand*" indicates the single- or double-stranded DNA that passes through the structure with variable length.
- "*Adjuster module*" is the part that can control the length of the adjuster strand.
- "*Strut staples*" is used to describe the staples that can bind to the adjuster strand. It replaces "*adjuster strut*", "*strut staples*", and "*adjuster staples*".
- "*Included angle*" is used throughout the manuscript instead of "*angle*" or "*bending angle*".

(2) Specify in fig. 1b (or in a supplementary figure) how exactly a rigid unit is converted into a hinge. Is it correct that staples are omitted or do in addition staples have to be exchanged?

► Basically, a hinge module can be made by eliminating all staples comprising a structure module. If one needs to reinforce the bending rigidity of the hinge module, some additional staples can be inserted in the hinge module (refer to Fig. 5a or Supplementary Fig. 1). We added sentences to explain it in the manuscript.

“To make a flexible hinge module having rotational degree of freedom, we can simply eliminate the existing staples constituting the structure module while maintaining scaffold strands at the cross-section (Fig. 1b), instead of reducing the number of DNA helices as in the previous studies. The bending stiffness of the hinge module can be tuned by controlling the number of dsDNA and staple crossovers inside the hinge module (Supplementary Fig. 1).”

(3) The language should sometimes be more precise, for example some adjectives are seem misplaced.

- line 22 *“unprecedented efficiency”* sounds a bit bold and it is not clear what exactly is meant. How about *...can provide a powerful, user-friendly and cost-effective design approach... or something similar?*

- l. 26: What is an *“extensive design domain”* or l. 28 an *“escalating demand”*?

- l. 37: No, the sequence of the scaffold strand never changes. A different segment is hybridized to the adjuster module, right?

- l. 51: what is meant by *sequence-independent region*?

► We made some modifications as follows.

- *“unprecedented efficiency in the design of”* is changed to *“a versatile and cost-effective procedure in the design of”* in order to avoid ambiguity.

- We added an explanation and revised the words.

“The extensive design space of scaffolded DNA origami comes from the availability of drawing a unique scaffold pathway with corresponding sequence design of staple strands for each structure.”

“Also, there has been many attempts to construct dynamic structures (...)”

- Yes, the reviewer’s point is correct. We revised that sentence to avoid any confusion as follows.

“A modular design method using two-dimensional repeating scaffold pathway to create two- and three-dimensional structures was only recently reported”

- We intended to express that the scaffold pathway in each module region is preserved, but the expression *“sequence-independent”* may induce ambiguity. Therefore, we changed the sentence as follows.

“Our modular design method starts from partitioning the structure by drawing a periodic scaffold path filling the cross-section.”

(4) Title: The title could focus more on the modularity, which is in this reviewer’s opinion the main selling point.

► We fully agree on the reviewer's opinion. In order to highlight the modularity, we changed the title to *'Polymorphic design of DNA origami structures through mechanical control of modular components'*.

(5) l. 41: ref. 27 actually also reported 3D structures, not only 2D, but that approach is not suitable for the mechanical control Lee et al. show here.

► We wanted to indicate that, in ref. 27, the modular scaffold pathway, not the structures themselves made by the design method there, was restricted in 2D. To clarify, we added the following statement to the main text.

"A modular design method using two-dimensional repeating scaffold pathway to create two- and three-dimensional structures was only recently reported"

(6) Past and present tense is mixed up repeatedly, e.g. in the results or l. 242 (was instead of is changed)

► We carefully and thoroughly proofread and revise the manuscript.

(7) There should always be a space between number and unit (often nt is used without space in text and figures)

► We checked and corrected this issue in the manuscript as well as in supporting information.

(8) Figure 2: delete "~" in front of numbers. Is 12.8% replacement not accurate? Is replacement actually correct or are some staple strands just omitted?

► In Figure 2, "~" was used to indicate the range of the portion of replaced staples. We added minimum portion of replaced staples for each design group. Detailed number of replaced staples for each design can be found in Supplementary Note.

Reviewers' Comments:

Reviewer #1:

Remarks to the Author:

The authors replied satisfactorily to most of the questions. However, with regard to the yields it is not enough just to give the yield (which should btw not be given with decimals).

A definition of what successful bending is should be given in the supporting information preferably with an AFM image where successfully and unsuccessfully bent structures are indicated.

Furthermore, the number of structures counted should be included to qualify how representative the yield is.

When this has been addressed I recommend acceptance.

Reviewer #2:

Remarks to the Author:

The authors have improved their manuscript where necessary and it could be published now.

Response to the referees' comments

Our point-by-point response to the reviewers' comments follows:

Reviewer #1's comments to the authors:

(1) The authors replied satisfactorily to most of the questions. However, with regard to the yields it is not enough just to give the yield (which should btw not be given with decimals).

A definition of what successful bending is should be given in the supporting information preferably with an AFM image where successfully and unsuccessfully bent structures are indicated.

► We provided detailed explanations on the definition of structural folding yield and the process of its calculation with representative cases in Supplementary Fig. 17 as shown below. Also, the exact number of analyzed monomer and well-folded structures are summarized in Supplementary Table 2 in addition to the percentage yield values. We think the provided information is sufficient to present quantitative structural folding yield of the designs we demonstrated.

(Supplementary Fig. 17) (...) Well-folded structure should have all hinges bent towards proper direction and amount (shown as orange circles), in accordance with the schematic design. Misfolded structures have either less number of bent region or have hinge(s) bent to opposite direction.

(2) Furthermore, the number of structures counted should be included to qualify how representative the yield is.

► In order to calculate the structural folding yield, we analyzed more than 250 individual monomer structures for each design. As described in our response above, the number of analyzed structures and well-folded structures are listed in Supplementary Table 2.

Reviewer #2's comment to the authors:

(1) The authors have improved their manuscript where necessary and it could be published now.

► We appreciate the reviewer's commitment.